# Antiviral Shrimp lncRNA06 Possesses Anti-Tumor Activity by Inducing Apoptosis of Human Gastric Cancer Stem Cells in a Cross-Species Manner

**DOI:** 10.3390/md22050221

**Published:** 2024-05-15

**Authors:** Ahmad Khan, Anas Mohammed, Xiaobo Zhang

**Affiliations:** 1College of Life Sciences, Laboratory for Marine Biology and Biotechnology of Pilot National Laboratory for Marine Science and Technology (Qingdao), Southern Marine Science and Engineering Guangdong Laboratory (Zhuhai), Zhejiang University, Hangzhou 310058, China; ahmadqau19@yahoo.com (A.K.); anas_elzein@yahoo.com (A.M.); 2Government Post Graduate College Miran Shah, Miran Shah 28200, Pakistan

**Keywords:** antiviral lncRNA, shrimp, human gastric cancer stem cells, tumorigenesis, cross-species regulation

## Abstract

Virus infection causes the metabolic disorder of host cells, whereas the metabolic disorder of cells is one of the major causes of tumorigenesis, suggesting that antiviral molecules might possess anti-tumor activities by regulating cell metabolism. As the key regulators of gene expression, long non-coding RNAs (lncRNAs) play vital roles in the regulation of cell metabolism. However, the influence of antiviral lncRNAs on tumorigenesis has not been explored. To address this issue, the antiviral and anti-tumor capacities of shrimp lncRNAs were characterized in this study. The results revealed that shrimp lncRNA06, having antiviral activity in shrimp, could suppress the tumorigenesis of human gastric cancer stem cells (GCSCs) via triggering apoptosis of GCSCs in a cross-species manner. Shrimp lncRNA06 could sponge human miR-17-5p to suppress the stemness of GCSCs via the miR-17-5p-p21 axis. At the same time, shrimp lncRNA06 could bind to ATP synthase subunit beta (ATP5F1B) to enhance the stability of the ATP5F1B protein in GCSCs, thus suppressing the tumorigenesis of GCSCs. The in vivo data demonstrated that shrimp lncRNA06 promoted apoptosis and inhibited the stemness of GCSCs through interactions with ATP5F1B and miR-17-5p, leading to the suppression of the tumorigenesis of GCSCs. Therefore, our findings highlighted that antiviral lncRNAs possessed anti-tumor capacities and that antiviral lncRNAs could be the anti-tumor reservoir for the treatment of human cancers.

## 1. Introduction

Virus infection in host cells is caused by viral and host factors and the surrounding environment. The host microbiota may enhance viral survival by improving virion stability and boosting coinfection rates [1,2,3]. The virus finishes its life cycle in host cells, and thus it depends on the metabolic mechanisms of living cells [4,5]. To achieve the maximum required energy and substance for the virus’s life cycle, the virus controls the metabolic homeostasis of the host cells, such as glycolysis and the pentose phosphate system [6]. Therefore, virus infection must result in the metabolic disturbance of the affected cells [5,7]. Upon sensing the invading viruses, the host cells generate some antiviral molecules, such as non-coding RNAs, secondary metabolites, and proteins, to defend against virus infection [7]. These antiviral molecules may block aberrant metabolism, restore metabolic balance, and ultimately fight against the proliferation of the virus. In essence, the antiviral molecules are to maintain the metabolic homeostasis of host cells [7]. Tumorigenesis is a complex process that includes genetic, environmental, and cellular mechanisms, of which the metabolic disorder of cells is one of the main causes of cancer [7]. In the aspect of metabolic disorder, cancer cells and virus-infected cells have the same signature [7]. Because the nature of antiviral molecules is to maintain the metabolic homeostasis of cells, the antiviral molecules may possess anti-tumor activity. The metabolic disorders of cells are commonly correlated with the regulation of gene expression patterns [7]. In virus-infected cells and tumor cells, the dysregulation of genes occurs. The accumulated evidence has shown that non-coding RNAs (ncRNAs), such as microRNAs (miRNAs) and long noncoding RNAs (lncRNAs), play essential roles in the regulation of gene expressions [7,8]. In this context, the antiviral non-coding RNAs, generated during the stress response of host cells to virus infection, may possess anti-tumor activity by maintaining the homeostasis of cells in a gene-expression-regulation manner [7]. A miRNA has several targets, suggesting that a single miRNA can control the expression of many genes from various animal species. As reported, shrimp miR-S8, shrimp miR-35, and shrimp miR-34 can simultaneously target shrimp genes and human genes, thus showing the antiviral capacity in shrimp and anti-tumor activity in humans in a cross-species manner [5,6,9,10,11]. At present, however, the influence of lncRNAs on antiviral and anti-tumor activities in animals and humans has not been characterized.

LncRNAs, the non-coding RNAs longer than two hundred nucleotides [12], serve critical regulatory roles in a wide range of illnesses, particularly malignant tumors [13]. In both biological processes and carcinogenesis, lncRNAs perform crucial regulatory functions of gene expression via interacting with proteins and/or miRNAs [14,15]. Many lncRNAs have been identified as competing endogenous RNAs (ceRNA), which behave as “miRNA sponges” by inhibiting specific miRNAs from interacting with target mRNAs to control gene expression [16,17]. LHFPL3-AS1-long exists for retaining the stemness of melanoma stem cells by directly interacting with miR-181 to prevent the mRNA degradation of its target gene Bcl-2 [18]. In gastric cancer, LINC00922 increases gastric cancer development by targeting miR-204-5p [19]. Some lncRNAs can bind to proteins to regulate gene expression [20]. LINC00152 carries out its intended action by attaching to the epidermal growth factor receptor (EGFR) to activate the EGFR-mediated pathway, leading to the carcinogenesis of gastric cancer [21]. LncRNAs have been shown to have both cis and trans functions [8]. The cis-acting lncRNAs can affect gene expression in an allele-specific manner because of the genomic homology among the targets, lncRNAs, and DNA elements that have been transcribed to control the expression of the adjacent genes [8,18,22]. By interacting with tissue-specific chromatin modifications, such as DNA methyltransferases and histone-modifying complexes, lncRNAs trans-regulate the transcription of genes [23,24,25]. Although lncRNAs have attracted more and more attention, the cross-species regulation of lncRNAs has not been extensively investigated.

To address the cross-species regulation of lncRNAs and their underlying mechanisms, the upregulated lncRNAs in the shrimp challenged with white spot syndrome virus (WSSV), a virus which could infect crustaceans, were evaluated and then the lncRNAs possessing antiviral activity in shrimp and anti-tumor capacity in humans were characterized in this study. The results revealed that a shrimp lncRNA (lncRNA06), having antiviral activity in shrimp, could suppress tumorigenesis in gastric cancer stem cells by the miR-17-5p-p21 axis.

## 2. Results

### 2.1. Antiviral Activity of Shrimp lncRNA06

To explore the antiviral lncRNAs in shrimp, shrimp were challenged by WSSV, followed by the evaluation of the expression profiles of shrimp lncRNAs. Based on the sequencing data (Bioproject NCBI Accession PRJNA932984), seven lncRNAs were significantly upregulated in the WSSV-challenged shrimp (Figure 1A), suggesting that these lncRNAs might play important roles in shrimp antiviral immunity. Based on the screening of seven lncRNAs, the silencing of lncRNA06 significantly affected the virus infection in shrimp. Therefore, lncRNA06 was further characterized.

The results of quantitative real-time PCR and Northern blot showed that lncRNA06 was significantly upregulated in the WSSV-challenged shrimp compared with the control (healthy shrimp) (Figure 1B), suggesting that shrimp lncRNA06 had effects on virus infection.

To reveal the role of shrimp lncRNA06 in the virus–host interaction, lncRNA06 was knocked down or overexpressed in the WSSV-infected shrimp, followed by the evaluation of WSSV infection. The results of quantitative real-time PCR showed that lncRNA06 was silenced in shrimp compared to the control (Figure 1C). The lncRNA06 silencing led to significant increases in WSSV copies in shrimp and shrimp mortality compared to the controls (WSSV alone and WSSV + lncRNA06-siRNA-scrambled) (Figure 1D,E), indicating that lncRNA06 played a negative role in virus infection. On the other hand, when lncRNA06 was overexpressed in shrimp (Figure 1F), the WSSV copies in shrimp and the shrimp cumulative mortality were significantly decreased compared to the controls (Figure 1G,H). These data demonstrated that lncRNA06 could suppress virus infection in shrimp.

Collectively, these findings revealed that shrimp lncRNA06 possessed antiviral activity in shrimp.

### 2.2. Effects of Shrimp lncRNA06 on Human Gastric Cancer Stem Cells

To examine the influence of shrimp lncRNA06 on human tumors, the cancer stem cells sorted from gastric cancer, melanoma, breast cancer, or liver cancer cell lines were transfected with shrimp lncRNA06, and then the cell viability was examined. Among the four types of cancer, shrimp lncRNA06 could only affect the viability of gastric cancer stem cells (Figure 2A). Thus, the influence of shrimp lncRNA06 on gastric cancer stem cells (GCSCs) was further characterized.

To explore the role of shrimp lncRNA06 in the tumorigenesis of human gastric cancer, GCSCs sorted from HGC-27 (GCSC-HGC) and MKN-45 (GCSC-MKN) were transfected with shrimp lncRNA06, followed by the examination of cell viability. The results showed that shrimp lncRNA06 was expressed in GCSCs (Figure 2B). The expression of lncRNA06 significantly decreased the viability of GCSCs (Figure 2C), indicating that shrimp lncRNA06 could suppress the proliferation of GCSCs. The results of cell cycle assays revealed that lncRNA06 led to cell cycle arrest of GCSCs in the G0/G1 phase (Figure 2D). The lncRNA06-mediated cell cycle arrest of GCSCs further triggered the apoptosis of GCSCs (Figure 2E,F). These data demonstrated that shrimp lncRNA06 could suppress the proliferation of GCSCs by inducing cell cycle arrest and further apoptosis.

To evaluate the influence of shrimp lncRNA06 on the stemness of GCSCs, lncRNA06 was stably expressed in GCSCs, and then the tumorsphere formation capacity of GCSCs was examined. The results revealed that shrimp lncRNA06 significantly decreased the tumorsphere formation percentage of GCSCs compared with the control (Figure 2G). At the same time, the expression levels of stemness genes in lncRNA06-treated GCSCs were significantly decreased (Figure 2H,I). These data indicated that shrimp lncRNA06 could inhibit the tumorigenesis of GCSCs.

Taken together, these data demonstrated that shrimp lncRNA06 could suppress tumorigenesis of GCSCs via inducing cell cycle arrest and apoptosis of GCSCs, indicating that the antiviral lncRNA06 in shrimp possessed anti-tumor activity in human beings.

### 2.3. Underlying Mechanism of Shrimp lncRNA06 in GCSCs

To reveal the mechanism of shrimp lncRNA06 in GCSCs, the human miRNAs that interacted with shrimp lncRNA06 were predicted. Based on the prediction, three miRNAs (miR-17-5p, miR-93, and miR-106b) were the potential targets of lncRNA06 (Figure 3A). To explore the interaction between lncRNA06 and miRNAs, dual luciferase assays were conducted. The results indicated that the luciferase activity of GCSCs (GCSC-HGC and GCSC-MKN) co-transfected with miR-17-5p and lncRNA06 was significantly decreased compared to the control (Figure 3B). However, the luciferase activity of GCSCs co-transfected with miR-93 or miR-106b and lncRNA06 was comparable to the control (Figure 3B). These data showed the direct interaction between miR-17-5p and shrimp lncRNA06.

To reveal the role of miR-17-5p in GCSCs, the expression profile of miR-17-5p in GCSCs and GCNCCs (gastric cancer non-stem cells) was determined. The results revealed that miR-17-5p was significantly upregulated in GCSCs compared with GCNSCs (Figure 3C), suggesting that miR-17-5p played an important role in GCSCs. To determine the influence of miR-17-5p on the properties of GCSCs, miR-17-5p was knocked down in GCSCs (Figure 3D). The silencing of miR-17-5p significantly reduced the viability of GCSCs compared with the control (Figure 3E). The miR-17-5p-mediated suppression of GCSCs’ viability resulted from the cell cycle arrest in the S phase (Figure 3F). The cell cycle arrest led to the apoptosis of miR-17-5p-silencing GCSCs (Figure 3G,H). At the same time, the results showed that miR-17-5p silencing significantly downregulated the expression of stemness genes (Figure 3I). The percentage of tumorsphere formation of miR-17-5p-silenced GCSCs was significantly reduced compared to the control (Figure 3J). These results demonstrated that miR-17-5p played a positive role in the proliferation and stemness of GCSCs.

To explore the mechanism of miR-17-5p in GCSCs, the target genes of miR-17-5p were predicted. The results indicated that CDKN1A (cyclin-dependent kinase inhibitor 1A or p21) and TP53INP1 (tumor protein p53 inducible nuclear protein 1) were the potential target genes of miR-17-5p (Figure 3K). To evaluate the interaction between miR-17-5p and p21 or TP53INP1, miR-17-5p was overexpressed in GCSCs. The results revealed that the miR-17-5p overexpression led to a significantly decreased p21 expression level, but it had no effect on TP53INP1 expression (Figure 3L,M), indicating that p21 was the target gene of miR-17-5p. To determine the direct interaction between p21 and miR-17-5p, dual luciferase assays were performed. The results indicated that the luciferase activity of the cells co-transfected with miR-17-5p and p21 was significantly decreased compared to the control, while the luciferase activity of the cells co-transfected with miR-17-5p and p21-mutant was not changed (Figure 3N), showing the direct interaction between miR-17-5p and p21. These data indicated that p21 was the target gene of miR-17-5p.

To assess the impact of shrimp lncRNA06 on p21, the expression level of p21 was examined in GCSCs transfected with lncRNA06. The results revealed that shrimp lncRNA06 significantly upregulated p21 in GCSCs (Figure 3O). When GCSCs were co-transfected with lncRNA06 and miR-175p, the expression profile of p21 was comparable to that of the control (Figure 3O). These data demonstrated that shrimp lncRNA06 promoted p21 expression via targeting miR-175p in GCSCs.

Collectively, these findings revealed that shrimp lncRNA06 could bind to human miR-17-5p to suppress the interaction between miR-17-5p and its target gene p21, leading to the upregulation of p21 in GCSCs (Figure 3P).

### 2.4. Role of p21 in GCSCs

To evaluate and explore the influence of p21 on GCSCs, the role of p21 in GCSCs was characterized. The results revealed that p21 was significantly downregulated in GCSCs compared with GCNSCs (Figure 4A). Thus, p21 was overexpressed in GCSCs, followed by the examination of cell properties. The results showed that p21 was overexpressed in GCSCs (Figure 4B). The overexpression of p21 resulted in the suppression of GCSCs’ viability (Figure 4C), indicating that p21 overexpression could inhibit the proliferation of GCSCs. The suppression of GCSCs’ proliferation resulted from cell cycle arrest in the G0/G1 phase (Figure 4D). The cell cycle arrest of GCSCs triggered the apoptosis of GCSCs (Figure 4E,F). These data demonstrated that p21 played a negative role in GCSCs.

To assess the effects of p21 overexpression on the stemness of GCSCs, the capability of tumorsphere formation and stemness gene expression of p21-overexpressed GCSCs were characterized. The results indicated that p21 overexpression led to a significantly decreased tumorsphere formation capacity of GCSCs compared with the controls (Figure 4G). At the same time, the expression levels of stemness genes in p21-overexpressed GCSCs were significantly reduced (Figure 4H). These results revealed that p21 could suppress the tumorigenesis of GCSCs.

Taken together, these results show that p21 functions as a tumor suppressor by inducing the apoptosis of GCSCs and inhibiting their stemness.

### 2.5. Influence of lncRNA06-Protein Interaction on GCSCs

To explore the proteins that interacted with shrimp lncRNA06 in GCSCs, an RNA pulldown assay was performed. The results demonstrated that a specific protein was bound to lncRNA06 (Figure 5A). The protein was identified as ATP synthase subunit beta (ATP5F1B) (Figure 5A). Western blot analysis confirmed the mass spectrometric identification (Figure 5B). These data indicated that shrimp lncRNA06 could interact with the human ATP5F1B protein.

To further characterize the direct interaction between shrimp lncRNA06 and human ATP5F1B protein, an EMSA analysis was performed. The results showed that lncRNA06 was bound to the ATP5F1B protein in a concentration-dependent manner (Figure 5C). To determine the sites of shrimp lncRNA06 interacting with the ATP5F1B protein, the truncated fragments of lncRNA06 were constructed based on the secondary structure of lncRNA06. The data from RNA pull-down experiments demonstrated that a 533-bp fragment (nucleotides from 532 to 1065 bp) of lncRNA06 was bound to the ATP5F1B protein (Figure 5D). These results revealed that shrimp lncRNA06 could interact with human ATP5F1B protein in a cross-species manner.

To evaluate the influence of shrimp lncRNA06 on the stability of ATP5F1B protein, GCSCs were transfected with lncRNA06, and then the mRNA and protein levels of ATP5F1B were examined. The results indicated that the mRNA level of ATP5F1B was not affected by shrimp lncRNA06, while the ATP5F1B protein level was significantly increased in the lncRNA06-transfected GCSCs (Figure 5E). These data demonstrated that shrimp lncRNA06 could enhance the stability of the ATP5F1B protein in GCSCs.

As reported, ATP5F1B, a novel RNA-binding protein, is involved in the proliferation and metastasis of gastric cancer cells via the ATP-P2X7-FAK/AKT/MMP2 axis [26]. Therefore, these findings revealed that shrimp lncRNA06 could increase the stability of the ATP5F1B protein in GCSCs to suppress tumorigenesis of GCSCs.

### 2.6. Role of Shrimp lncRNA06 in Tumorigenesis of GCSCs In Vivo

To assess the effects of shrimp lncRNA06 on gastric tumor progression in vivo, GCSCs transfected with shrimp lncRNA06 or lncRNA06-scrambled were subcutaneously injected into non-obese diabetes/severe combined immunodeficiency (NOD/SCID) mice, followed by tumor examination (Figure 6A). The results showed that the tumor volume of the mice injected with lncRNA06-expressing GCSCs was significantly decreased compared with that of the control mice (Figure 6B), indicating that shrimp lncRNA06 prevented tumor growth in mice. The sizes and weights of the solid tumors yielded essentially similar results (Figure 6C,D). These data revealed that shrimp lncRNA06 could suppress the tumorigenesis of GCSCs in vivo.

To evaluate whether the inhibition of tumorigenesis of GCSCs in mice resulted from the lncRNA06-miR-17-5p-p21 axis, the expression level of p21 in the solid tumors of the mice inoculated with the shrimp lncRNA06 or lncRNA06-scrambled-transfected GCSCs was examined. The results showed that lncRNA06 dramatically increased the p21 expression level (Figure 6E). At the same time, the data from immunohistochemical analysis revealed that the expression of p21, the targeting gene of miR-17-5p, was significantly increased in the solid tumors of the mice transfected with lncRNA06-expressing GCSCs compared with the control (Figure 6F). These results indicated that shrimp lncRNA06 prevented tumor growth in vivo via the lncRNA06-miR-17-5p-p21 pathway.

To explore the influence of shrimp lncRNA06 on the proliferation of GCSCs in vivo, the expression level of the proliferative gene ki67 was examined by immunohistochemical analysis. The results demonstrated that ki67 was downregulated in the solid tumors of the mice transfected with the lncRNA06-expressing GCSCs compared with the control mice (Figure 6G). These data indicated that shrimp lncRNA06 could suppress the proliferation of GCSCs in vivo.

Taken together, shrimp lncRNA06 could bind to the ATP5F1B protein to increase its stability and interact with miR-17-5p to inhibit the p21 mRNA degradation mediated by the interaction between miR-17-5p and p21 mRNA (Figure 6H). As a result, the interactions between lncRNA06 and ATP5F1B as well as miR-17-5p and p21 promoted apoptosis and suppressed the proliferation of GCSCs, leading to the suppression of tumorigenesis of GCSCs (Figure 6H).

## 3. Discussion

Gastric cancer characteristics, such as tumor initiation and progression, resistance to chemotherapy, and recurrence of the disease, are related to a tumor subpopulation called gastric cancer stem cells (GCSCs) [27]. GCSCs with infinite potential for self-regeneration, differentiation, and tumor regeneration play significant roles in gastric cancer’s refractory properties [11,28]. Because of their high capability for proliferation and stemness, GCSCs seem to be promising therapeutic targets for gastric cancer [11,28]. Cancer stem cells continuously exhibit a substantial level of genes associated with their stemness for maintaining their cancer stem cell capabilities [29]. LncRNAs, critical regulatory elements of gene expression, have important roles in the pluripotency, differentiation, self-renewal, and tumorigenicity of cancer stem cells via regulating the expression of transcriptional genes and oncogenic signaling pathways [30,31]. In this study, the findings revealed that the antiviral shrimp lncRNA06 possessed anti-tumor capacity against the tumorigenesis of human GCSCs in a cross-species manner. The influence of shrimp lncRNA06 on four types of cancer stem cells, including gastric cancer, melanoma, breast cancer, and liver cancer stem cells, was characterized. However, shrimp lncRNA06 could only affect the viability of GCSCs. In the future, more cancers could be used to characterize the effects of shrimp lncRNA06. As reported, the shrimp antiviral miRNAs have anti-tumor capacities in humans via regulating their target genes’ expressions [5,6,9,10,11]. The antiviral miRNAs produced in the host cell stress response to virus infection in shrimp possess anti-tumor effects by retaining the cell’s metabolic homeostasis [7]. In a cross-species approach, shrimp miR-S8, miR-35, and miR-34 can simultaneously target shrimp genes and human genes, demonstrating the antiviral potential in shrimp and the anti-tumor activity in humans [5,6,9,10,11]. In this context, our findings showed that non-coding RNAs, including miRNAs and lncRNAs, having antiviral activity might possess anti-tumor capacity in humans. Animal non-coding RNAs with antiviral activity could serve as significant reserves for the development of anticancer drugs.

In this investigation, the findings revealed that shrimp lncRNA06 could suppress tumorigenesis of GCSCs in a cross-species manner via two strategies: acting as a sponge of human miR-17-5p and binding to human ATP5F1B protein. The interaction between shrimp lncRNA06 and human miR-17-5p inhibited the degradation of p21 mediated by miR-17-5p in GCSCs. The shrimp lncRNA06-miR-17-5p-p21 axis triggered the apoptosis of GCSCs and suppressed the stemness of GCSCs. In gastric cancer, miR-17-5p, the prominent member of the miR-17-92 cluster, regulates stem cell characteristics to promote the growth of gastric cancer cells via specifically targeting runt-related transcription factor 3 (RUNX3) gene, while the knockdown of miR-17-5p results in a significant reduction of proliferation, invasion, and metastasis via targeting phosphatase and tensin homolog deleted on chromosome 10 (PTEN) [32,33,34]. P21, a member of the Cip/Kip family, acts as a regulator of multiple tumor suppressor pathways for anti-proliferative activities via mediating biological activities primarily by binding to and inhibiting the kinase activity of the cyclin-dependent kinases (CDKs) [35,36]. In recurrent ovarian cancer patients, a p53–p21 signature of cancer stemness is found [37]. The binding of p21 to NF-κB and STAT inhibits the expression of anti-apoptotic proteins such as BCL-2, c-FLIP, BCL-XL, and XIAP, hence triggering apoptosis [38]. In this context, our study presented a novel mechanism of animal lncRNA via downregulating human miR-17-5p to upregulate human p21, leading to the inhibition of tumorigenesis in a cross-species manner. Except for the shrimp lncRNA06-miR-17-5p-p21 axis, our findings revealed that shrimp lncRNA06 could bind to human ATP5F1B protein to enhance protein stability, thus inhibiting the tumorigenesis of GCSCs. ATP5F1B, an RNA-binding protein, can enhance the proliferative and metastatic capacities of gastric cancer cells through the ATP-P2X7-FAK/AKT/MMP2 pathway [26]. The downstream effectors of ATP5F1B in GCSCs could be confirmed in future work. Therefore, our investigation contributed novel insights into the mechanisms of shrimp lncRNA06 in the tumorigenesis of human gastric cancer. Based on our findings, more shrimp antiviral lncRNAs merited to be explored to obtain anti-tumor lncRNAs for the treatment of human gastric cancers. In the future, the influence of epigenetic changes or post-transcriptional modifications on the expression and function of shrimp lncRNA06 in GCSCs merits further characterization. The potential off-target effects of shrimp lncRNA06 on the cellular processes of GCSCs could also be evaluated in further investigations. Additionally, the challenges of the utilization of shrimp lncRNA06 in clinics, such as the preclinical studies using the full-length shrimp incRNA06 or its binding sites for miR-17-5p and ATP5F1B and the development of effective delivery methods for lncRNA-based therapies in cancer, needed to be further investigated.

## 4. Materials and Methods

### 4.1. Shrimp Culture, Virus Infection, and Mortality Analysis

Shrimp (*Marsupenaeus japonicus*) with an average body weight of 6 to 8 g were cultured in tanks at 250 °C with air-pumped circulating seawater in groups containing 20 shrimp/group. Prior to virus infection into shrimp, three shrimp per group were selected randomly, and then subjected to PCR using WSSV-specific primers (5′-TATTGTCTCTCCTGACGTAC-3′ and 5′-CACATTCTT CACGAGTCTAC-3′) to ensure that the shrimp used in the experiments were WSSV-free. The virus-free shrimp were infected with WSSV, and at various time points post-infection, the shrimp hemolymph was collected for later use. The cumulative shrimp mortality was examined every day.

### 4.2. Shrimp lncRNA Sequencing and Data Analysis

Total RNA was extracted from the hemocytes of healthy shrimp and WSSV-challenged shrimp using Trizol reagent (Invitrogen, Carlsbad, CA, USA) in accordance with the manufacturer’s instructions. The extracted RNAs were then subjected to lncRNA sequencing on an illumina Novaseq™ 6000 (LC-Bio Technology Co., Ltd., Hangzhou, China) following the recommended protocol (Appendix A). The sequences were mapped to the transcriptome of shrimp using gffcompare (https://github.com/gpertea/gffcompare/, accessed on 1 September 2015). Two computational algorithms, CNCI1 and CPC (Coding Potential Calculator), were utilized to predict lncRNAs associated with shrimp.

### 4.3. Northern Blot

Total RNA was extracted from cells or tissues using the Pure Cell/Bacteria Kit for RNAprep (Tiangen Biotech, Beijing, China). After separation by gel electrophoresis, the RNAs were transferred to a nylon membrane (Amersham Biosciences, Buckinghamshire, UK). The membrane was cross-linked with ultraviolet and then pre-hybridized at 42 °C in DIG (digoxigenin) Easy Hyb granule buffer (Roche, Schweiz, Switzerland) for 1.5 h. The hybridized DIG-labeled probe for lncRNA06 (5′-TATTATGCCATCCTCATCAAGCCA-3′) or control U6 probe (5′-GGGC CATGCTAA TCTTCTCTGTATCGTT-3′) was incubated with the membrane overnight at 42 °C. The signal detection was performed using the Starter Kit II DIG High Prime DNA Labeling and Detection (Roche, Schweiz, Switzerland).

### 4.4. Shrimp lncRNA06 Silencing or Overexpression in Shrimp

To knock down the shrimp lncRNA06 expression in shrimp, the sequence-specific siRNA (lncRNA06-siRNA, 5′-GCCAUCUCCUAGUUGUAUATT-3′ and 5′-UAUACAACUAGG AGAUGGCTT-3′), synthesized by GenePharma Co., Ltd. (Shanghai, China), was injected into shrimp at 15 μg/shrimp. As a control, lncRNA06-siRNA-scrambled (5′-UUCUCCGAA CGUGUCACGUTT-3′ and 5′-ACGUGACACGUUCGGAGAATT-3′) was included in the injection. Twelve hours later, the shrimp were re-injected with siRNA again. At different times after the last injection, three shrimp, randomly selected from each group, were collected for later use.

To perform overexpression of lncRNA06, the full-length lncRNA06 was amplified using lncRNA06-specific primers (5′-CGCTGGGAAATCTCTCTTG-3′ and 5′-AGGACTGACAA TAGTGTTGGG-3′) and synthesized with a T7 kit for in vitro transcription (TaKaRa, Ostu, Japan) according to the protocol of the manufacturer. Shrimp were injected with the synthesized lncRNA06 or lncRNA06-scrambled at 15 μg/shrimp. Twelve hours later, the shrimp were re-injected with the same lncRNA. At different time points after the last injection, three shrimp were randomly selected from each group and collected for later use.

### 4.5. Quantitative Real-Time PCR

Total RNA was extracted from shrimp hemocytes using an RNA isolation kit (Ambion, Austin, TX, USA). The complementary DNA was synthesized with a reverse transcription system (Toyobo, Osaka, Japan) following the instructions of the manufacturer. Quantitative real-time PCR was performed using SYBR Green PCR Master Mix (Vazyme Biotech Corporation, Nanjing, China) with sequence-specific primers [lncRNA06, 5′-GCTTTGATGAGGATGGC A-3′ and 5′-CTCCAGGTAAGTTCAGTCCAG-3′; shrimp U6, 5′-TTCACGAATTTGCGTG TCAT-3′ and 5′-CGCTTCGGCAGCACATATAC-3′; OCT4 (octamer-binding transcription factor 4), 5′-GCCGCTGGCTTATAGA AGGT-3′ and 5′-GGAGCTTGGCAAATTGCTCG-3′; SOX2 (SRY-box transcription factor 2), 5′-AGTTACGCGCACATGAACGG-3′ and 5′-CTCTCCTCTTTTGCACCCCT-3′; ALDH1 (aldehyde dehydrogenase 1), 5′-CAAGATCC AGGGCCGTACAA-3′ and 5′-GTCCAAGTCGGCATCAGCTA-3′; p21, 5′-GTCAGTTCC TTGTGGAGCCG-3′ and 5′-CTGCCTCC TCCCAACTCATC-3′); tumor protein p53 inducible nuclear protein 1 (TP53INP1), 5′-TGGGCCTTCTATCTTGGATG-3′ and 5′-GCAAGGCTGACACCACTGTA-3′; ATP synthase subunit beta (ATP5F1B), 5′-TGCCCCT GCTACTACGTTTG-3′ and 5′-ACCTCAGCAACCTGGAATGG-3′; glyceraldehyde-3-phosphate dehydrogenase (GAPDH), 5′-GGTATCGTGGAAGGACTCATGAC-3′ and 5′-A TGCCAGTGAGCTTCCCGTTCAG-3′; miR-17-5p, 5′-GCGCAAAGTGCTTACAGTGC-3′ and 5′-AGTGCAG GGTCCGAGGT ATT-3′; U6, 5′-CTCGCTTCGGCAGCACA-3′ and 5′-AACGCTTCACGA ATTTGCGT-3′)].

To examine the WSSV copies, the viral genome was extracted from WSSV-infected shrimp using an isolation kit for DNA (Omega, Norcross, GA, USA). The quantitative real-time PCR was accomplished with the WSSV-specific primers (5′-TTGGTTTCATGCCC GAGATT-3′ and 5′-CCTTGGTCAGCCCCCTTGA-3′) and the TaqMan fluorescence-based probe (5′-FAM-TGCTGCCGTCCTCCAATAMRA-3). The PCR mixture (up to 10 μL) contained 5 μL of the Premix Ex Taq (TaKaRa, Ostu, Japan), roundabout 200 ng of the sample DNA template along with 0.2 μL of 10 μM of each primer, and 0.2 μL of 10 μM of the TaqMan fluorogenic probe at 0.2 μM final concentration. A plasmid containing a 1400 bp WSSV genomic DNA fragment was used as an internal standard [39]. The PCR condition was 95 °C 1 min, followed by 45 cycles of 95 °C for 30 s, 52 °C for 30 s, and 72 °C for 30 s [40].

### 4.6. Cell Culture

Cancer stem cells were sorted from HGC-27, MKN-45, HepG2, MDA-MB-435, and MCF-7 cell lines in our laboratory previously [28,41,42]. Cancer stem cells were cultured in serum-free DMEM/F-12 medium (Gibco, New York, NY, USA) supplemented with 2% B-27 (Sigma, St. Louis, MO, USA), a 100 U mixture of the pen/strep (Shijiazhuang Pharmaceutical Group Co., Ltd., Shijiazhuang, China), 20 ng/mL epidermal growth factor, 10 ng/mL essential fibroblast growth factor, and 5 μg/mL insulin (Beyotime Biotechnology, Shanghai, China). The cells were cultured at 37 °C in a humidified atmosphere with 5% CO_2_.

### 4.7. Expression of Shrimp lncRNA06 in Cells

Shrimp lncRNA06 was amplified by PCR using sequence-specific primers (5′-CGGATC CCGCTGGGAAATCTCTCTTG-3′ and 5′-CGGAATTCAGGACTGACAATAGTGTTGG G-3′) and then cloned into the pcDNA3.1 plasmid (Promega, Madison, WI, USA). Cells (1 × 10^5^ per mL) were transfected with the recombinant plasmid expressing lncRNA06 or vector alone using Lipofectamine 2000 (Thermo Fisher Scientific, Waltham, MA, USA). At different times after transfection, the cells were collected for later use.

### 4.8. Cell Viability Assay

Cell viability assays were conducted using MTS [3-(4, 5-dimethylthiazol-2-yl)-5-(3-carboxymethoxyphenyl)-2-(4-sulfophenyl)-2H-tetrazolium, inner salt] (Promega, Madison, WI, USA) according to the manufacturer’s protocol. Briefly, 20 μL of CellTiter 96^®^ AQueous One Solution Reagent was added to the cells and then incubated for 2 h at room temperature, followed by the measurement using the iMARKTM microplate reader at 490 nm (Bio-Rad, Hercules, CA, USA).

### 4.9. Cell Cycle Assay

The cell cycle assay was performed using FACS (fluorescence-activated cell sorting). Cells were washed three times with cold PBS and then incubated with 70% ethanol at 20 °C overnight. After fixation, the cells were centrifuged at 300× *g* for 10 min. Then, the cells were resuspended in PBS and incubated with RNase A (Sangon Biotech, Toronto, ON, Canada) for 30 min at 37 °C in the dark. The cells were stained with propidium iodide (PI) (Sigma-Aldrich, St. Louis, MO, USA) at 37 °C for 15 min. Subsequently, the cells were examined using the FACSCalibur flow cytometer (BD Biosciences, San Jose, CA, USA).

### 4.10. Analysis of Caspase 3/7 Activity

The caspase 3/7 activity of cells was determined using the caspase-Glo 3/7 kit (Promega, Madison, WI, USA) following the manufacturer’s protocol. Cells at 1 × 10^4^/well were placed in a 96-well plate. The caspase-Glo 3/7 reagent (Promega, Madison, WI, USA) was added into every well. After incubation for 1 h in the dark at room temperature, the cell luminescence was measured using a microplate reader (Promega, Madison, WI, USA).

### 4.11. Apoptosis Detection by Annexin V

Apoptosis analysis was conducted according to the manufacturer’s instructions using the fluorescein isothiocyanate (FITC)-annexin V apoptosis detection kit I (BD Biosciences, San Jose, CA, USA). Cells were rinsed with cold PBS. Then, the cells were incubated with FITC-annexin V and PI (propidium iodide) at room temperature for 15 min in the dark, followed by the addition of 1×annexin binding buffer. The apoptotic cells were examined using flow cytometry (BD Biosciences, San Jose, CA, USA).

### 4.12. Tumorsphere Formation Assay

Cells were suspended in DMEM/F-12 medium (Gibco, New York, NY, USA) to carry out tumorsphere formation assays in a non-adherent and serum-free environment. A single cell was grown in an ultralow adherent 96-well plate containing serum-free DMEM/F-12 (Gibco, New York, NY, USA) medium supplemented with 2% B-27 (Sigma, St. Louis, MO, USA), a 100 U mixture of the pen/strep (Shijiazhuang Pharmaceutical Group Co., Ltd., Shijiazhuang, China), 20 ng/mL epidermal growth factor, 10 ng/mL essential fibroblast growth factor, and 5 μg/mL insulin factor (Beyotime Biotechnology, Shanghai, China). The cells were observed every day under a light microscope.

### 4.13. Western Blot

Proteins were separated by 10% SDS-PAGE (SDS-polyacrylamide gel electrophoresis) and then electrotransferred onto a nitrocellulose membrane (GE Healthcare, Waukesha, WI, USA) in the transferring buffer (25 mM Tris-HCl, 190 mM glycine, 25% methyl alcohol). After blocking with 5% non-fat milk in the TBST buffer (20 mM Tris-HCl, 150 mM NaCl, 0.05% Tween-20, pH 8.0) at room temperature for 60 min, the membrane was incubated with a primary antibody at room temperature overnight. Then, the membrane was rinsed with PBS and incubated with the horseradish peroxidase (HRP)-conjugated secondary antibody (BioRad, Hercules, CA, USA) for 3 h at room temperature. The signals of the membrane were detected using the Western Lightning Plus-ECL package (Perkin Elmer, Waltham, MA, USA).

### 4.14. Prediction of Human miRNAs Targeted by Shrimp lncRNA06 and Target Genes of miRNAs

The human miRNAs targeted by shrimp lncRNA06 were predicted using miRDB (https://mirdb.org/custom.html) and RNA22 v2 (https://cm.jefferson.edu/rna22/) algorithms accessed on 10 March 2023. The overlapped miRNAs were the potential targets of shrimp lncRNA06.

The putative target genes of miRNAs were predicted using TargetScan, miRanda, and PicTar algorithms accessed on 10 March 2023. The overlapped genes were considered the targets of miRNAs.

### 4.15. Dual-Luciferase Reporter Assay

Shrimp lncRNA06 was cloned into the luciferase reporter vector pmirGLO (Promega, Madison, WI, USA) using sequence-specific primers (5′-CTCGAGTGGATGGAAGCTATTTCTGACACC AAATGCACTTTAATGT-3′ and 5′-TCTAGAACATTAAAGTGCATTTGGTGTCAGAAA TAGCTTCCATCCACTCGAG-3′). As a control, lncRNA06 was mutated and then cloned into pmirGLO using sequence-specific primers (5′-CTCGAGTGGATGGAAGCTATTTCTG ACACCAAATATCAGGGAATGT-3′ and 5′-TCTAGAACATTCCCTGATATTTGGTGTC AGAAATAGCTTCCATCCACTCGAG-3′). The recombinant plasmid and one of the synthesized miRNAs (miR-17-5p, miR-93, and miR-106b) were co-transfected into GCSCs using Lipofectamine 2000 (Thermo Fisher, Waltham, MA, USA). At 48 h after transfection, the firefly and renilla luciferase activities were measured using the dual luciferase reporter assay system (Promega, Madison, WI, USA) according to the manufacturer’s protocol.

### 4.16. Silencing or Overexpression of miRNAs or Genes in Cells

To silence a miRNA, cells (1 × 10^5^ per mL) were transfected with 50 nM of anti-miRNA oligonucleotide (AMO) (AMO-miR-17-5p, 5′-CUACCUGCACUGUAAGCACUUUG-3′) or AMO-miR-17-5p-scrambled (5′-CAGUACUUUUGUGUAGUACAA-3′) using Lipofectamine 2000 (Thermo Fisher Scientific, Waltham, MA, USA). The AMO-miRNA was synthesized by GenePharma Co., Ltd., Shanghai, China). At 48 h after transfection, the cells were collected for later use.

To overexpress a miRNA, cells (1 × 10^5^ per mL) were cultured in a six-well plate and transfected with 50 nM of the synthesized miRNA (GenePharma Co., Ltd., Shanghai, China) using Lipofectamine 2000 (Thermo Fisher Scientific, Waltham, MA, USA). A total of 48 h after transfection, the cells were collected for later use.

To overexpress p21, p21 was cloned into the pcDNA3.1 plasmid (Invitrogen, Carlsbad, CA, USA) using the sequence-specific primers (5′-CGGGATCCTGCCGAAGTCAGTTCCTTGT-3′ and 5′-C GTCTAGAGCACCTGCTGTATATTCAGC-3′). Then, GCSCs (1 × 10^5^ per mL) were transfected with the recombinant pcDNA3.1 expressing p21 or plasmid alone using Lipofectamine 2000 (Thermo Fisher Scientific, Waltham, MA, USA) according to the manufacturer’s instructions. At different times after transfection, the cells were collected for later use.

### 4.17. Dual Luciferase Activity Assay

To conduct the dual luciferase activity assay, two recombinant plasmids were constructed using sequence-specific primers (p21, 5′-CTCGAGTCCCTCCCCAGTTCATTGCACTTTG -3′, and 5′-TCTAGACAAAGTGCAATGAACTGGGGAGGGA-3′; p21-mutant 5′-CTCG AGTCCCTCCCCAGTTCATTTGATGGGG-3′ and 5′-TCTAGACCCCATCAAA TGAAC TGGGGAGGGA-3′). The binding sites for miR-17-5p within the 3′-UTR of p21 were mutated from 5′-TCCCTCCCCAGTTCATTGCACTTTG-3 to 5′-TCCCTCCCCAGTTCAT TTGATGGGG-3′. The p21 and p21-mutant were cloned into the pmirGLO dual-luciferase miRNA target expression vector (Promega, Madison, WI, USA). Subsequently, 50 nM of the synthesized miR-17-5p (5′-CAAAGUGCUUACAGUGCAGGUAG-3′) or control miRNA (5′-UUCUCCGAACGUGU CACGUTT-3′) was co-transfected with 2000 ng of the plasmid expressing p21 or p21-mutant into GCSCs using Lipofectamine 2000. At 48 h after transfection, the luciferase activity of the cells was examined according to the manufacturer’s instructions (Promega, Madison, WI, USA).

### 4.18. RNA Pulldown Assay

The biotin-labeled shrimp lncRNA06 (5′-TAAGTCTTCTTTTCTTTGTTTGTCTTTTGT-biotin-TEG-3′) was synthesized by Hangzhou Youkang Biotechnology Inc., China. Streptavidin magnetic beads (Beyotime Biotechnology, Shanghai, China) were washed with 1 × TBS (Beyotime Biotechnology, Shanghai, China) and then with 0.05 M NaCl. The biotin-lncRNA06 was incubated with streptavidin magnetic beads (Beyotime Biotechnology, Shanghai, China) in the binding and washing buffer I (2×) (10 mM Tris-HCl, 1 mM EDTA, 2 M NaCl, 0.01%–0.1% Tween-20, pH 7.5) for 30 min at room temperature, followed by the addition of the cell lysate. The mixture was incubated for 2 h at 4 °C. After washing with the binding and washing buffer, the proteins were eluted using 0.1% sodium dodecyl sulfate (SDS). The eluted proteins were separated by sodium dodecyl sulfate-polyacrylamide gel electrophoresis (SDS-PAGE) and stained with Coomassie brilliant blue (Beytime Biotechnology, Shanghai, China). The proteins were identified using mass spectrometry.

### 4.19. Tumorigenesis of GCSCs in Mice

To assess the effects of shrimp lncRNA06 on tumorigenesis of GCSCs in vivo, GCSCs were transfected with shrimp lncRNA06 or lncRNA06-scrambled and cultured for 48 h. The cells, resuspended with physiological saline, were mixed with the Matrigel (Becton, Mountain View, CA, USA) at a ratio of 2:1 and then subcutaneously injected into 5 non-obese diabetes/severe combined immunodeficiency (NOD/SCID) female mice of age 6 to 8 weeks old and body weight less than 18 g (1 × 10^5^ cells/per mouse). The tumor volume was examined every 5 days. Six weeks later, the mice were sacrificed, and the solid tumors were collected for later use. All the animal experiments were performed according to the instructions approved by the China Institutional Animal Care and Use Committee (IACUC).

### 4.20. Immunohistochemical Analysis

The solid tumors of mice were cut into 5-μm-thick sections and then loaded onto the precoated slides with 10% 3-triethoxysilylpropylamine (Merck, Darmstadt, Germany). The slides were socked within Xylol for 1 h and washed with a series of decreasing concentrations of alcohol (100, 95, and 80%). After deparaffinizing, a microwave antigen retrieval of the sections was performed in TEC buffer (0.05 M ethylenediaminetetraacetic acid, 0.05 M Tris-HCl, 0.02 M Na-citrate, pH 7.8) for 5 min, followed by blocking with peroxidase. The slides were incubated for 12 h with a primer antibody in a humified chamber and subsequently incubated with the biotinylated secondary antibody (Vector, Grunberg, Germany) for 30 min. Subsequently, the slides were stained with diaminobenzidine (Sigma, St. Louis, MO, USA) for 10 min at room temperature to label proteins and counterstained with hematoxylin for nucleic labeling.

### 4.21. Electrophoretic Mobility Shift Assay

The electrophoretic mobility shift assay (EMSA) was performed to examine the interaction between lncRNA and protein. The recombinant protein at different concentrations was incubated with 50 mM synthesized lncRNA in EMSA buffer (Beyotime Biotechnology, Shanghai, China) at 37 °C for 30 min. As a control, lncRNA alone was included in the assays. The mixture was subsequently separated by 1% agarose gel electrophoresis and stained with ethidium bromide to detect RNAs. The protein used was detected by SDS-PAGE with Coomassie brilliant blue staining.

### 4.22. Prediction of lncRNA Secondary Structure

The secondary structure of shrimp lncRNA06 was predicted using the online RNAfold (http://rna.tbi.univie.ac.at/cgi-bin/RNAWebSuite/RNAfold.cgi) accessed on 20 June 2023.

### 4.23. Statistical Analysis

All numerical data in this study were presented in the form of a mean value accompanied by the standard deviation, denoting the variability within the dataset. To determine the statistical significance of differences between various treatments, one-way analysis of variance (ANOVA) and Student’s *t*-test were employed. To ensure the reliability and consistency of the results, all experimental assays were conducted in triplicate at the biological level.

## 5. Conclusions

The antiviral non-coding RNAs, generated during the stress response of host cells to virus infection, may possess anti-tumor activity by maintaining the homeostasis of cells in a gene-expression-regulation manner. Our findings revealed that shrimp lncRNA06 could suppress the tumorigenesis of GCSCs in a cross-species manner via two strategies: acting as a sponge of human miR-17-5p and binding to the human ATP5F1B protein. The interaction between shrimp lncRNA06 and human miR-17-5p inhibited the degradation of p21 mediated by miR-17-5p in GCSCs. The shrimp lncRNA06-miR-17-5p-p21 axis triggered the apoptosis of GCSCs and suppressed the stemness of GCSCs. Shrimp lncRNA06 could bind to the human ATP5F1B protein to enhance protein stability, thus inhibiting the tumorigenesis of GCSCs. Therefore, our findings highlighted that antiviral lncRNAs possessed anti-tumor capacities and that animal lncRNAs could be the anti-tumor reservoir for the treatment of human cancers.

## Figures and Tables

**Figure 1 marinedrugs-22-00221-f001:**
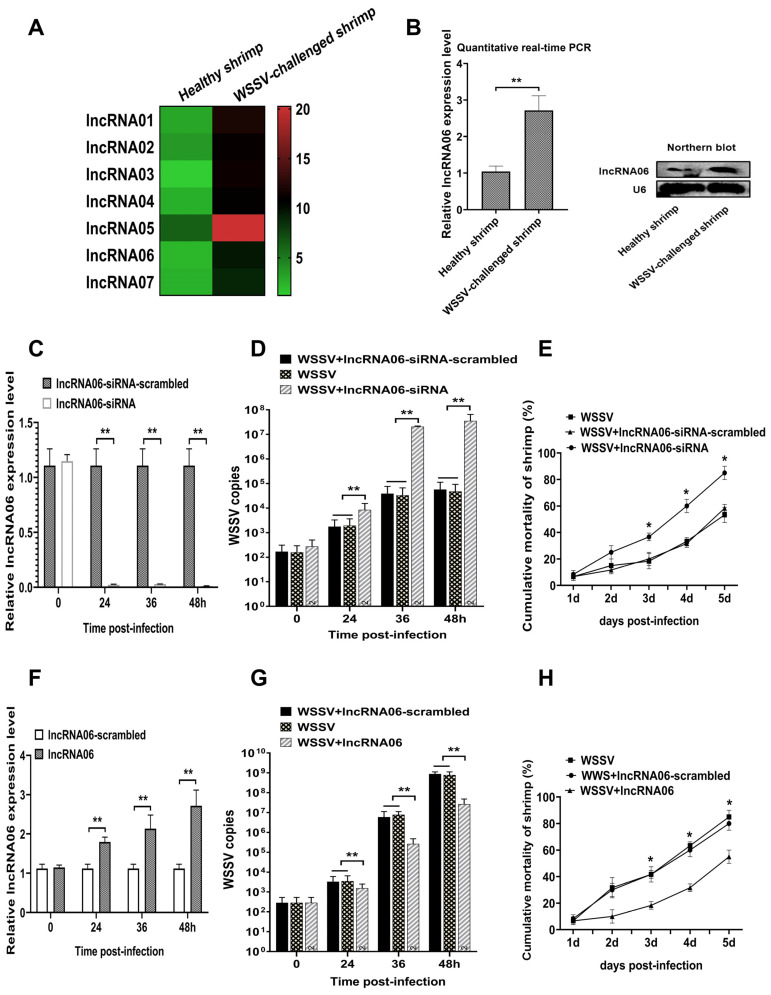
Antiviral activity of shrimp lncRNA06. (**A**) The upregulated lncRNAs in the WSSV-challenged shrimp. Shrimp were infected with WSSV. At 48 h post-infection, shrimp hemocytes were collected and subjected to lncRNA sequencing. The heatmap showed the upregulated lncRNAs in the WSSV-challenged shrimp. (**B**) Upregulation of lncRNA06 in virus-infected shrimp. At 48 h post-infection, the expression level of lncRNA06 in shrimp hemocytes was determined using quantitative real-time PCR (**, *p* < 0.01) and Northern blot. U6 was used as a control. (**C**) Silencing of lncRNA06 in shrimp. Shrimp were injected with lncRNA06-siRNA or lncRNA06-siRNA-scrambled. At different times after injection, shrimp hemocytes were collected to examine the lncRNA06 expression using quantitative real-time PCR (**, *p* < 0.01). U6 was used as a control. (**D**) Effects of lncRNA06 silencing on virus infection. The synthesized lncRNA06-siRNA and WSSV were co-injected into shrimp. At different time points post-infection, WSSV copies in hemocytes were detected (**, *p* < 0.01). (**E**) Influence of lncRNA06 silencing on shrimp mortality. The cumulative mortality of shrimp was monitored at different times (*, *p* < 0.05). (**F**) Overexpression of lncRNA06 in shrimp. The synthesized lncRNA06 was injected into shrimp. At different times after injection, the expression level of lncRNA06 was examined with quantitative real-time PCR. U6 was used as a control (**, *p* < 0.01). (**G**) Impact of lncRNA06 overexpression on WSSV infection. The WSSV copies of the virus-challenged shrimp overexpressing lncRNA06 were determined (**, *p* < 0.01). (**H**) Influence of lncRNA06 overexpression on shrimp mortality. The cumulative mortality of shrimp was examined every day (*, *p* < 0.05).

**Figure 2 marinedrugs-22-00221-f002:**
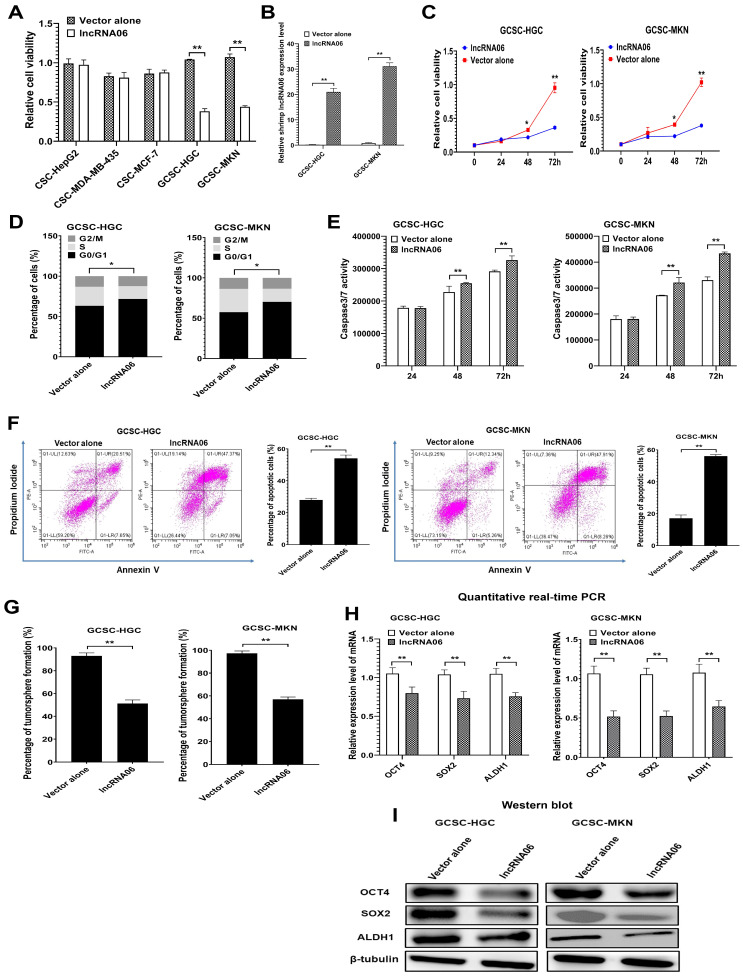
Effects of shrimp lncRNA06 on human gastric cancer stem cells. (**A**) Influence of shrimp lncRNA06 on the viability of cancer stem cells. Cancer stem cells were transfected with shrimp lncRNA06. The vector alone was used as a control (control). Forty-eight hours later, the cell viability was assessed (**, *p* < 0.01). (**B**) Expression of shrimp lncRNA06 in GCSCs. At 48 h after transfection, the expression level of lncRNA06 in GCSCs was analyzed by quantitative real-time PCR (**, *p* < 0.01). (**C**) Impact of shrimp lncRNA06 on the viability of GCSCs. At different times after transfection, the cell viability of GCSCs was examined (*, *p* < 0.05; **, *p* < 0.01). (**D**) Role of shrimp lncRNA06 in the cell cycle. At 48 h after transfection, the cell cycle was examined using flow cytometry (*, *p* < 0.05). (**E**) Detection of caspase 3/7 activity. The caspase 3/7 activity of GCSCs transfected with shrimp lncRNA06 or vector alone was determined (**, *p* < 0.01). (**F**) Detection of apoptosis using the annexin V assay. GCSCs were examined using flow cytometry at 48 h after transfection (**, *p* < 0.01). (**G**) Influence of shrimp lncRNA06 on the tumorsphere formation capacity of GCSCs. The lncRNA06-transfected GCSCs were subjected to tumorsphere formation assays. Seven days later, the percentage of tumorsphere formation was evaluated (**, *p* < 0.01). (**H**) Effects of shrimp lncRNA06 on the expression of stemness genes in GCSCs. The expression profiles of stemness genes in GCSCs were examined at 48 h after transfection (**, *p* < 0.01). (**I**) Western blot analysis of stemness genes in GCSCs. β-tubulin was used as a control.

**Figure 3 marinedrugs-22-00221-f003:**
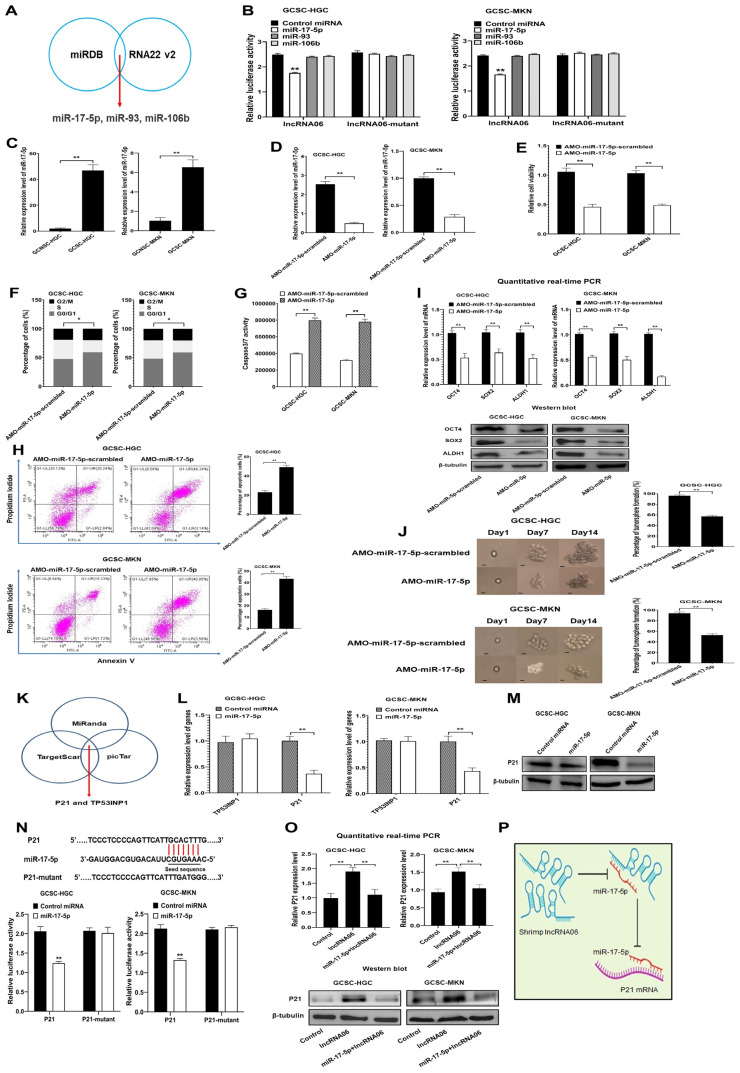
The underlying mechanism of shrimp lncRNA06 in GCSCs. (**A**) Prediction of miRNAs interacting with shrimp lncRNA06. The prediction was performed, and the overlapped miRNAs were the potential targets of lncRNA06. (**B**) Direct interaction between miRNAs and shrimp lncRNA06. GCSCs were co-transfected with shrimp lncRNA06 and a miRNA. Forty-eight hours later, the firefly and renilla luciferase activities of the cells were examined (**, *p* < 0.01). (**C**) Expression of miR-17-5p in GCSCs and GCNSCs. The expression level of miR-17-5p was examined (**, *p* < 0.01). (**D**) Knockdown of miR-17-5p in GCSCs. GCSCs were transfected with AMO-miR-17-5p or AMO-miR-17-5p-scrambled. At 48 h after transfection, the expression of miR-17-5p was determined (**, *p* < 0.01). (**E**) Impact of miR-17-5p silencing on cell viability. At 48 h after transfection, the cell viability of AMO-transfected GCSCs was determined (**, *p* < 0.01). (**F**) Effects of miR-17-5p silencing on the cell cycle. At 48 h after transfection, the cell cycle of GCSCs transfected with AMO was examined using flow cytometry (*, *p* < 0.05). (**G**) Effects of miR-17-5p downregulation on the apoptosis of GCSCs. GCSCs transfected with AMO were subjected to caspase 3/7 activity detection at 48 h after transfection (**, *p* < 0.01). (**H**) Detection of apoptosis using annexin V assays. At 48 h after transfection, apoptosis of GCSCs was examined using annexin V assay (**, *p* < 0.01). (**I**) Influence of miR-17-5p silencing on the expression of stemness genes in GCSCs. At 48 h after transfection of AMO, the cells were subjected to quantitative real-time PCR (**, *p* < 0.01) and Western blot to examine the expression of stemness genes. (**J**) Influence of miR-17-5p silencing on the ability of tumorsphere formation in GCSCs. Tumorsphere formation of a single cell was shown on the left. The percentage of tumorsphere formation was examined at day 14 after transfection (**, *p* < 0.01). Scale bar: 10 μm. (**K**) The potential genes targeted by miR-17-5p. The overlapped genes were the potential targets of miR-17-5p. (**L**) Impact of miR-17-5p overexpression on the expression of target genes. At 48 h after transfection of miRNA, the gene expression was examined using quantitative real-time PCR (**, *p* < 0.01). (**M**) Western blot analysis of p21 in the miR-17-5p-overexpressed GCSCs. β-tubulin was used as a control. (**N**) Direct interaction between miR-17-5p and p21. GCSCs were co-transfected with miR-17-5p or control miRNA and p21 or p21-mutant. Forty-eight hours later, the firefly and renilla luciferase activities were examined (**, *p* < 0.01). (**O**) Influence of shrimp lncRNA06 on the expression of p21 in GCSCs. At 48 h after transfection, the expression level of p21 was determined using quantitative real-time PCR (**, *p* < 0.01) and Western blot. The cells without any treatment were used as controls. (**P**) Model for the underlying mechanism of shrimp lncRNA06 in GCSCs.

**Figure 4 marinedrugs-22-00221-f004:**
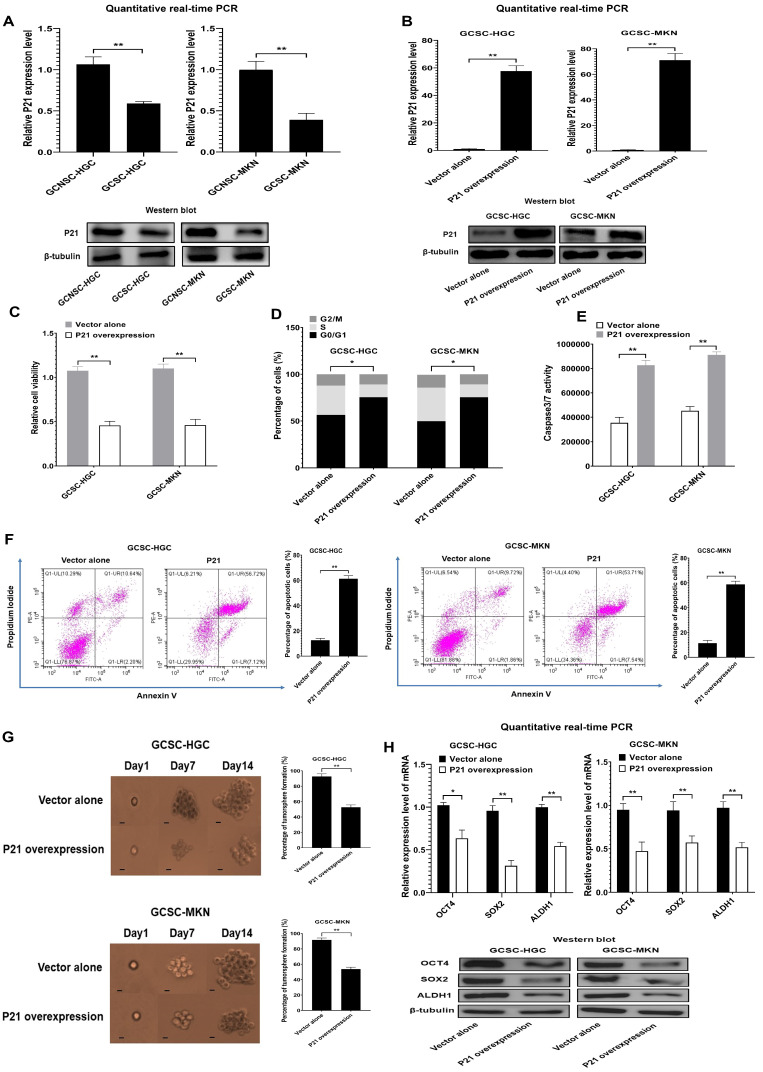
Role of p21 in GCSCs. (**A**) Differential expression of p21 in GCSCs and GCNSCs. The expression level of p21 was examined using quantitative real-time PCR (**, *p* < 0.01) and Western blot. β-tubulin was used as a control. (**B**) Overexpression of p21 in GCSCs. GCSCs were transfected with the recombinant plasmid expressing p21. At 48 h after transfection, the cells were subjected to quantitative real-time PCR (**, *p* < 0.01) and Western blot. (**C**) Influence of p21 overexpression on cell viability. GCSCs overexpressing p21 were subjected to cell viability assays at 48 h after transfection (**, *p* < 0.01). (**D**) Effects of p21 overexpression on the cell cycle. The cell cycle of p21-overexpressing GCSCs was examined at 48 h after transfection (*, *p* < 0.05). (**E**) Impact of p21 overexpression on apoptosis of GCSCs. At 48 h after transfection, the caspase 3/7 activity of GCSCs overexpressing p21 was determined (**, *p* < 0.01). (**F**) Detection of apoptosis using the annexin V assay. GCSCs overexpressing p21 were subjected to annexin V assays at 48 h after treatment (**, *p* < 0.01). (**G**) Influence of p21 overexpression on the ability of tumorsphere formation in GCSCs. GCSCs overexpressing p21 were subjected to tumorsphere-forming assays. Fourteen days later, the percentage of tumorsphere formation was evaluated. Scale bar: 10 μm (**, *p* < 0.01). (**H**) Expression of stemness genes in p21-overexpressed GCSCs. The expression of stemness genes in GCSCs overexpressing p21 were examined at 48 h after transfection using quantitative real-time PCR (*, *p* < 0.05; **, *p* < 0.01) and Western blot.

**Figure 5 marinedrugs-22-00221-f005:**
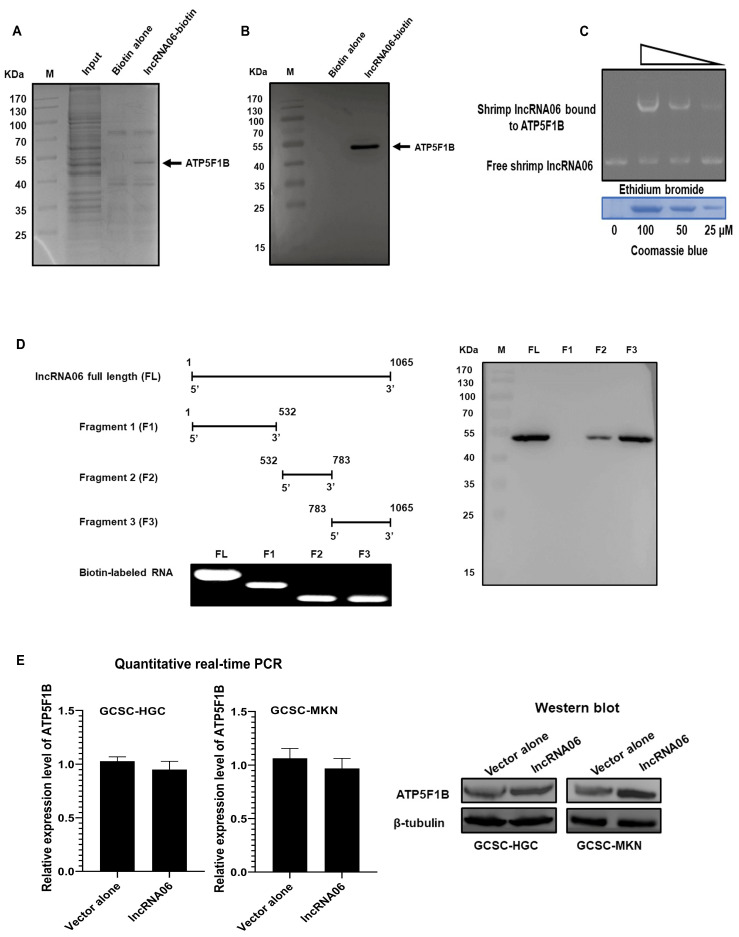
Influence of lncRNA06-protein interaction on GCSCs. (**A**) The protein interacted with shrimp lncRNA06. The lysate of GCSCs was incubated with lncRNA06-coupled beads, followed by elution of the proteins. The proteins were identified using mass spectrometry. The protein identified was indicated with an arrow. M, protein marker. (**B**) Western blot analysis of the protein that interacted with shrimp lncRNA06. The eluted proteins of the lncRNA06 pull-down assays were examined using Western blot. M, protein marker. (**C**) Direct interaction between shrimp lncRNA06 and human ATP5F1B protein. Shrimp lncRNA06 was incubated with the recombinant ATP5F1B protein at different concentrations. The mixture was separated by agarose gel electrophoresis to detect RNAs (up). The protein used was separated by SDS-PAGE, followed by Coomassie brilliant blue staining (down). (**D**) The sites of shrimp lncRN06 engaging in interaction with the ATP5F1B protein. Shrimp lncRN06 was truncated (left) and then the biotin-labeled fragments were subjected to RNA pull-down assays using the lysate of GCSCs. The pulled-down proteins were analyzed by Western blot (right). (**E**) Impact of shrimp lncRNA06 on the stability of the ATP5F1B protein in GCSCs. At 48 h after transfection, the mRNA and protein levels of ATP5F1B in GCSCs were examined. β-tubulin was used as a control.

**Figure 6 marinedrugs-22-00221-f006:**
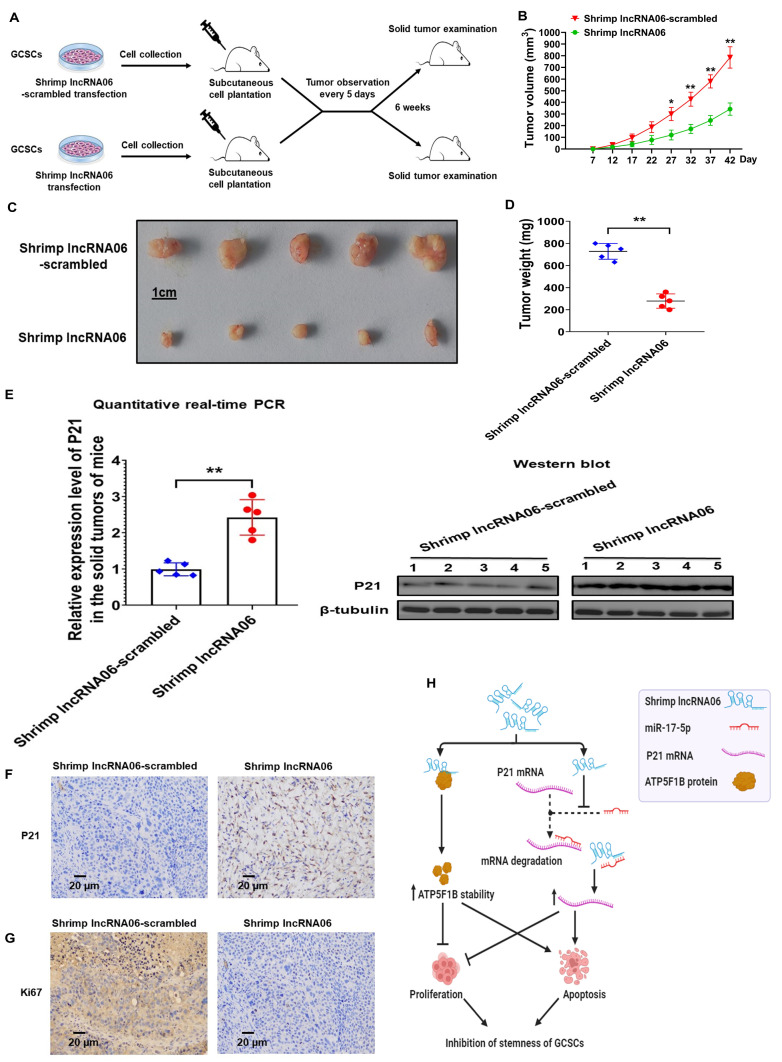
Role of shrimp lncRNA06 in the tumorigenesis of GCSCs in vivo. (**A**) Schematic diagram of tumorigenesis of GCSCs in NOD/SCID mice in vivo. GCSCs transfected with shrimp lncRNA06 were injected into NOD/SCID mice. The tumor volume was examined every 5 days, and the mice were sacrificed at week 6. (**B**) Evaluation of the solid tumor sizes resulting from mice receiving GCSCs transfected with shrimp lncRNA06. The horizontal axis indicated the days after cell inoculation into mice (*, *p* < 0.05; **, *p* < 0.01). (**C**) Effects of shrimp lncRNA06 on the tumor growth in mice. GCSCs expressing lncRNA06 were inoculated into mice. Six weeks later, the solid tumors of mice were examined. (**D**) Weight of the solid tumors of the mice inoculated with the lncRNA06-transfected GCSCs. The statistical significance of the difference between treatments was indicated with asterisks (**, *p* < 0.01). (**E**) The expression level of p21 in xenografts receiving GCSCs transfected with shrimp lncRNA06. The p21 expression was examined using quantitative real-time PCR (**, *p* < 0.01) or Western blot. β-tubulin was used as a control. (**F**) Immunohistochemical analysis of the expression of p21 in the solid tumor of mice. Brown represented the p21 protein, and blue represented the nuclei stained with hematoxylin. Scale bar: 20 μm. (**G**) Immunohistochemical analysis of the expression of Ki67 in the solid tumors of mice. Brown represented the Ki67 protein, and blue represented the nuclei. Scale bar: 20 μm. (**H**) Model for the underlying mechanism of shrimp lncRNA06 in tumorigenesis of GCSCs.

## Data Availability

The RNA-sequencing data that support the findings of this study are available in the National Center for Biotechnology Information (NCBI) with a (Bioproject NCBI Accession PRJNA932984). Most relevant data are included in the manuscript. Appendix A and additional raw data analyzed are available from the corresponding author on reasonable request.

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
