# Peer review of "Antiviral Shrimp lncRNA06 Possesses Anti-Tumor Activity by Inducing Apoptosis of Human Gastric Cancer Stem Cells in a Cross-Species Manner"

_marinedrugs, 2024, doi:10.3390/md22050221_

Round 1

Reviewer 1 Report

Comments and Suggestions for Authors

Please take a look at the comments I've attached.

Author Response

Dear Reviewers,

I am pleased to know that our manuscript entitled “Antiviral shrimp lncRNA06 possesses anti-tumor activity by inducing apoptosis of human gastric cancer stem cells in a cross-species manner” (Manuscript ID: marinedrugs-2938497) has the opportunity to be revised. I would like to take this opportunity to appreciate your efforts and to thank the anonymous reviewer for their valuable suggestions and comments. The manuscript has been subjected to revision accordingly. All changes made are highlighted in red in the revised manuscript. The particulars are listed below.

For Reviewer 1:

1) Question: The study demonstrates a novel function of shrimp lncRNA06 in suppressing the proliferation and stemness of human gastric cancer stem cells (GCSCs), which has important implications for developing new therapies for gastric cancer. What would be the potential effects of shrimp lncRNA06 on other types of cancer cells?

Answer: This is a good question. To examine the influence of shrimp lncRNA06 on human tumors, the cancer stem cells sorted from gastric cancer, melanoma, breast cancer and liver cancer cell lines were transfected with shrimp lncRNA06 and then the cell viability was examined. Among these four types of cancers, shrimp lncRNA06 could only affect the viability of gastric cancer stem cells (Fig 2A). Thus, the influence of shrimp lncRNA06 on gastric cancer stem cells (GCSCs) was further characterized in this study. This issue is discussed in the revised manuscript.

Revisions in the revised manuscript: lines 406-410.

2) Question: The study provides a detailed investigation of the underlying mechanism by which shrimp lncRNA06 regulates the proliferation and stemness of GCSCs, providing important insights into the molecular pathways involved in tumor growth and progression. The study investigates the interaction between lncRNA06 and three miRNAs (miR-17-5p, miR-93, and miR106b), but it is unclear whether other miRNAs may also be involved in regulating the effects of lncRNA06 on GCSCs.

Answer: Based on the prediction, 3 miRNAs (miR-17-5p, miR-93 and miR-106b) were the potential targets of lncRNA06. The data of the dual luciferase assays further indicated that there was a direct interaction between shrimp lncRNA06 and miR-17-5p (but not miR-93 and miR-106b). According to the comment, the manuscript is revised.

Revisions in the revised manuscript: lines 190-191.

3) Question: The study highlights the potential therapeutic value of lncRNA06 in treating gastric cancer, as it targets a key regulator of cell cycle progression and tumor growth. However, it would be essential to discuss the potential effects of other factors, such as epigenetic changes or post-transcriptional modifications, on the expression or function of lncRNA06 in GCSCs.

Answer: As suggested, the concern about the epigenetic changes or post-transcriptional modifications of shrimp lncRNA06 is discussed in the revised manuscript.

Revisions in the revised manuscript: lines 447-449.

4) Question: The study provides a comprehensive analysis of the role of p21 in GCSCs, demonstrating that it plays a critical role in regulating cell viability, proliferation, and stemness, which are important hallmarks of cancer. These observations were based on the effects of p21 overexpression on GCSCs and do not evaluate the effects of p21 knockdown or other manipulations that may influence p21 expression in GCSCs. Additionally, it would be essential to address the potential off-target effects of lncRNA06 overexpression on other cellular processes, which could affect the interpretation of the results.

Answer: We agree the reviewer’s comment. As shown in Fig 4A, p21 was significantly downregulated in GCSCs compared with GCNSCs. Therefore, p21 was overexpressed in GCSCs, followed by the examination of cell properties. According to the comments, the manuscript is modified.

Revisions in the revised manuscript: lines 267-270; lines 449-451.

5) Question: The study highlights the potential therapeutic value of p21 in treating gastric cancer, as it functions as a tumor suppressor by inducing apoptosis and inhibiting stemness. However, inserting a sentence to discuss the potential downstream effectors of p21 that may be involved in regulating apoptosis and stemness in GCSCs would strengthen the observation.

Answer: As suggested, the downstream effectors of p21 are described in the revised Discussion section.

Revisions in the revised manuscript: lines 434-436; References.

6) Question: The study identifies a novel interaction between shrimp lncRNA06 and human ATP5F1B protein, highlighting potential implications for cross-species interactions. However, evaluating the downstream effects of increased ATP5F1B protein on GCSCs would be critical. Also, the study demonstrates that shrimp lncRNA06 can enhance the stability of ATP5F1B protein in GCSCs, leading to increased tumorigenesis. It would be essential to provide insights into potential therapeutic strategies for targeting the shrimp lncRNA06 and ATP5F1B protein interaction in GCSCs.

Answer: It is known that ATP5F1B can promote the proliferation and metastasis of gastric cancer cells via the ATP-P2X7-FAK/AKT/MMP2 axis (Wang et al, 2021, The FASEB Journal, 35: 20649). In this study, the results revealed that shrimp lncRNA06 could increase the stability of ATP5F1B protein in GCSCs to suppress tumorigenesis. According to the comment, the manuscript is revised.

Revisions in the revised manuscript: lines 325-328; lines 443-444.

7) Question: The study provides evidence for the tumor-suppressive role of shrimp lncRNA06 in GCSCs in vivo, highlighting its potential as a therapeutic target for gastric cancer. It would be helpful to discuss its potential effects on other cell types or tissues, which is critical in generalizing the findings.

Answer: As suggested, it is discussed.

Revisions in the revised manuscript: lines 406-410.

8) Question: The study identifies the molecular mechanism underlying the tumor-suppressive effects of shrimp lncRNA06, including its interaction with ATP5F1B protein to increase its stability and its interaction with miR-17-5p to inhibit the degradation of p21 mRNA. This provides a comprehensive understanding of the lncRNA's role in tumorigenesis. However, the study does not explore the potential off-target effects of manipulating shrimp lncRNA06 expression, which could affect other cellular processes and gene expression patterns.

Answer: It is a good question. The potential off-target effect is discussed in the revised manuscript.

Revisions in the revised manuscript: lines 449-451.

9) Question: The discussion should acknowledge the study's limitations, such as using a xenograft mouse model and not evaluating the effects of manipulating shrimp lncRNA06 expression in other cell types or tissues. It would be essential to address the potential challenges in translating the findings to clinical applications, such as the need for further preclinical studies and the development of effective delivery methods for lncRNA-based therapies. Additionally, it focuses solely on the role of animal lncRNAs in tumorigenesis of human gastric cancer. It does not discuss the potential broader implications of the findings for other types of cancer or diseases.

Answer: According to the comment, the study's limitations are discussed.

Revisions in the revised manuscript: lines 451-453.

10) Question: The conclusion does not mention the study's limitations or the potential challenges in translating the findings to clinical applications, which were also not discussed in the discussion section. It overgeneralizes the findings by stating that "antiviral lncRNAs possessed anti-tumor capacities" without acknowledging the specificity of the function of shrimp lncRNA06 and the need for further research to determine whether other animal lncRNAs have similar antitumor properties. There are no suggestions for future research and directions for developing lncRNA-based therapies for human cancers

Answer: According to the comment, the discussion is modified.

Revisions in the revised manuscript: lines 445-447.

Thank you very much!

Sincerely yours,

Xiaobo Zhang

Reviewer 2 Report

Comments and Suggestions for Authors

See comments to Editor.

Also

This manuscript describes a huge amount of work on the cross-species antitumoral effects of shrimp lncRNA06. In this context the first question to be made is ¿What is exactly the rationale for mixing shrimp lncRNAs with human miRNAs and proteins? I think that the logical way of addressing this question would have been looking for the human homologue form of shrimp lncRNA06, even if there was only a partial homology, and using it on the experiments

There are many contradictions in the text,  in the abstract authors state that “shrimp lncRNA06, having antiviral activity in shrimp, could suppress tumorigenesis of human gastric cancer stem cells (GCSCs) (line 31)” but also that “p lncRNA06 could bind to ATP synthase subunit beta (ATP5F1B) to enhance the stability of ATP5F1B protein in GCSCs, thus promoting tumorigenesis of GCSCs (line 34)”

The first sentence of the abstract is, at least, surprising and inaccurate. Tumorigenesis is much more than a “metabolic disorder of cells” and the following suggestion is beyond current knowledge. Introduction is a complete nonsense. Do authors talk about a single virus or make general statements?

Comments on the Quality of English Language

Moderate editing required

Author Response

Dear Reviewers,

I am pleased to know that our manuscript entitled “Antiviral shrimp lncRNA06 possesses anti-tumor activity by inducing apoptosis of human gastric cancer stem cells in a cross-species manner” (Manuscript ID: marinedrugs-2938497) has the opportunity to be revised. I would like to take this opportunity to appreciate your efforts and to thank the anonymous reviewer for their valuable suggestions and comments. The manuscript has been subjected to revision accordingly. All changes made are highlighted in red in the revised manuscript. The particulars are listed below.

For Reviewer 2:

1) Question: What is exactly the rationale for mixing shrimp lncRNAs with human miRNAs and proteins? I think that the logical way of addressing this question would have been looking for the human homologue form of shrimp lncRNA06, even if there was only a partial homology, and using it on the experiments.

Answer: It is a good question. Because the nature of antiviral molecules is to maintain the metabolic homeostasis of cells, the antiviral molecules may possess anti-tumor activity. In this context, the antiviral non-coding RNAs, generated during the stress response of host cells to virus infection, may possess anti-tumor activity by maintaining the homeostasis of cells in a gene-expression-regulation manner. According to the comment, the manuscript is revised.

Revisions in the revised manuscript: lines 443-453.

2) Question: There are many contradictions in the text, in the abstract authors state that “shrimp lncRNA06, having antiviral activity in shrimp, could suppress tumorigenesis of human gastric cancer stem cells (GCSCs) (line 31)” but also that “p lncRNA06 could bind to ATP synthase subunit beta (ATP5F1B) to enhance the stability of ATP5F1B protein in GCSCs, thus promoting tumorigenesis of GCSCs (line 34)”

Answer: Sorry for our incorrect description. It is modified in the revised manuscript.

Revisions in the revised manuscript: lines 326-328.

3) Question: The first sentence of the abstract is, at least, surprising and inaccurate. Tumorigenesis is much more than a “metabolic disorder of cells” and the following suggestion is beyond current knowledge. Introduction is a complete nonsense. Do authors talk about a single virus or make general statements?

Answer: According to the comment, the manuscript is revised.

Revisions in the revised manuscript: lines 21-22; lines 53-56; lines 93-94.

4) Question: Comments on the Quality of English Language, Moderate editing required.

Answer: According to the comment, the manuscript has been proofread by an English speaker.

Revisions in the revised manuscript: the whole manuscript.

Thank you very much!

Sincerely yours,

Xiaobo Zhang

Reviewer 3 Report

Comments and Suggestions for Authors

The paper “Antiviral shrimp lncRNA06 possesses anti-tumor activity by inducing apoptosis of human gastric cancer stem cells in a cross-species manner” is interesting however, needs minor corrections before publication.

In general abbreviations should be added to all acronyms.

The figures need to be improved, most of them are blurred and is difficult to follow. All blots should be added completely to be observed properly.

Author Response

Dear Reviewers,

I am pleased to know that our manuscript entitled “Antiviral shrimp lncRNA06 possesses anti-tumor activity by inducing apoptosis of human gastric cancer stem cells in a cross-species manner” (Manuscript ID: marinedrugs-2938497) has the opportunity to be revised. I would like to take this opportunity to appreciate your efforts and to thank the anonymous reviewers for their valuable suggestions and comments. The manuscript has been subjected to revision accordingly. All changes made are highlighted in red in the revised manuscript. The particulars are listed below.

For Reviewer 3:

1) Question: In general abbreviations should be added to all acronyms.

Answer: As suggested, the abbreviations are listed in the revised manuscript.

Revisions in the revised manuscript: lines 727-742.

2) Question: The figures need to be improved, most of them are blurred and is difficult to follow. All blots should be added completely to be observed properly.

Answer: As suggested, the resolution of images is improved in the revised manuscript.

Revisions in the revised manuscript: Figs 1-6.

Thank you very much!

Sincerely yours,

Xiaobo Zhang

Round 2

Reviewer 1 Report

Comments and Suggestions for Authors

Recommended for publication.

Author Response

Thank you so much for considering our manuscript for publication.

Reviewer 2 Report

Comments and Suggestions for Authors

Although I acknowledge the work made by the authors to improve the manuscript, my original concern regarding the use of shrimp lncRNAs as antitumor agents in humans is still valid.

In order to facilitate the clinical translation of this work I would suggest the authors to determine the binding sites for miR-17-5p and ATP5F1B in lncRNA06 and repeat the experiments using these instead of the full-length lncRNA06.

¿What is the rationale for mapping the shrimp sequences to the human genome (M&M, lines 470-473)?. ¿Does this mean that there is an homologue sequence of lncRNA06 in humans? Even if partial, this data should be included in the manuscript. 

The way in which lncRNA06 was found is very poorly described. Much more details should be provided about sequencing results (a volcano plot would be ok), how were RNAs sequenced, etc.

Sequence of lncRNA06 SHOULD be deposited in the Genbank and a valid accession number should be provided. The one stated (PRJN932984) is not an accession number.

Comments on the Quality of English Language

Moderate editing required

Author Response

Dear reviewer 2,

I am pleased to know that our manuscript entitled “Antiviral shrimp lncRNA06 possesses anti-tumor activity by inducing apoptosis of human gastric cancer stem cells in a cross-species manner” (Manuscript ID: marinedrugs-2938497) has the opportunity to be re-revised. I would like to take this opportunity to appreciate your efforts and thanks for your valuable suggestions and comments. The manuscript has been subjected to re-revision accordingly. All changes made are highlighted in red in the revised manuscript. The particulars are listed in the attached file.

Round 3

Reviewer 2 Report

Comments and Suggestions for Authors

Only one comment. Authors must be aware that the Accn. number provided (PRJNA932984) is NOT a Genbank Accn. number but a BIOPROJECT  link at the NCBI. Please,instead of refering to "Genbank......." change to "Bioproject NCBI  Accession PRJNA932984 "